# Factors associated with suicidal ideation among medical residents in Tehran during the COVID-19 pandemic: A multicentric cross-sectional survey

Fahimeh Saeed[1], Elaheh Ghalehnovi[2], Mahdieh Saeidi[3], Neda Ali beigi[1], Mohsen Vahedi[4], Mohammadreza Shalbafan[5], Leila Kamalzadeh[6]*, Ali Nazeri Astaneh[1], Amir Hossein Jalali Nadoushan[5], Sheikh Shoib[1,7]

1 Psychosis Research Center, University of Social Welfare and Rehabilitation Sciences, Tehran, Iran, 2 Razi Vaccine and Serum research Institute, Karaj, Iran, 3 Research Center for Addiction and Risky Behaviors, Iran University of Medical Sciences, Tehran, Iran, 4 Department of Biostatistics and Epidemiology, Pediatric Neurorehabilitation Research Center, University of Social Welfare and Rehabilitation Sciences, Tehran, Iran, 5 Department of Psychiatry, Mental Health Research Center, Psychosocial Health Research Institute (PHRI), School of Medicine, Iran University of Medical Sciences, Tehran, Iran, 6 Department of Psychiatry, Geriatric mental health research center, School of Medicine, Iran University of Medical Sciences, Tehran, Iran, 7 Department of Health Services, Srinagar, Kashmir, India

* Kamalzadeh.I@iums.ac.ir, lkamalzadeh@gmail.com

**Data Availability Statement:** All relevant data are within the manuscript and its Supporting Information files.

## Abstract

### Background

The mental health of medical residents, challenged by their intensive training, is of utmost concern. In light of reported suicides among Iranian medical residents in 2021, this study investigates the factors behind suicidal ideation among medical residents during the COVID-19 pandemic in Tehran.

### Methods

This study conducted a cross-sectional online survey among medical residents in various specialties in Tehran, Iran, amidst the COVID-19 pandemic. Suicidal ideation was assessed using the Beck Scale for Suicidal Ideation (BSSI), while depression, anxiety, and stress were measured using the DASS-21. It also collected demographic and clinical data from the participants. The data were analyzed using descriptive statistics, the Chi-square test, and multiple linear regression to examine the prevalence and determinants of suicidal ideation among medical residents.

### Results

The study enrolled 353 medical residents and found that 34.3% of them had suicidal ideation, with 10.2% indicating a high risk. The study also found high levels of depression, anxiety, and stress among the participants. The variables that significantly predicted suicidal ideation were depression, history of alcohol/substance use, personal history of suicide attempts, history of self-mutilation, family history of suicide attempts, number of shifts in a

**Funding:** F.S. received funding support from the University of Social Welfare and Rehabilitation Sciences, Tehran, Iran, with grant number USWR1400-2714. The funder's website can be accessed at https://en.uswr.ac.ir/. The funders did not have any involvement in the study design, data collection and analysis, decision to publish, or preparation of the manuscript.

**Competing interests:** The authors have declared that no competing interests exist.

month, death of close persons because of COVID-19, and income. Depression was the strongest predictor of suicidal ideation.

## Conclusion

These findings underscore the urgent need for effective interventions and support systems to address the mental health needs of medical residents in Iran. The strategies should prioritize destigmatizing mental health, promoting access to mental health services, fostering a supportive training environment, and enhancing income opportunities.

## Introduction

The mental well-being of medical residents is a critical concern due to the demanding and challenging nature of their training, which can have profound implications for patient care [1]. Medical residency represents a pivotal phase in physicians' careers, characterized by long working hours, sleep deprivation, heightened stress levels, social deprivation, and exposure to emotionally taxing circumstances. These factors can cause considerable psychological distress and may increase the risk of developing psychiatric disorders [2].

Research conducted in low- and middle-income countries has shown a notable prevalence of depression and anxiety among residents in various specialized fields, ranging from 11% to 65% [3–7]. While residents in these countries face additional challenges, such as limited healthcare resources, low income, and disparities in healthcare distribution [4], it is crucial to recognize that high rates of mental health problems in medical residents are not exclusive to developing countries. Studies conducted in high-income countries have also reported similarly elevated rates of depressive and anxiety disorders among medical residents, ranging from 7% to 43% [8–11]. A comprehensive meta-analysis incorporating data from 54 studies worldwide revealed an overall prevalence of depressive symptoms of 28.8% among 17,560 resident physicians (95% CI, 25.3%-32.5%) [12]. These rates significantly exceed the lifetime prevalence of depression in the general adult population worldwide, which stands at 5%, as verified by the World Health Organization (WHO) [13].

Depression, particularly when comorbid with anxiety disorders, can lead to severe outcomes such as suicidal thoughts and behaviors [14]. Several studies have documented the occurrence of suicidal ideation among medical residents, with prevalence rates ranging from 4 to 35 percent [15–18]. According to the Accreditation Council for Graduate Medical Education (ACGME), suicide is the second leading cause of resident deaths in the USA, with the majority of suicides occurring during the first two years of training [19]. Disturbingly, reports from India indicate 105 suicide-related deaths among residents between 2010 and 2019, with academic stress and harassment identified as significant risk factors [20,21]. Various factors have been proposed to explain the high suicide rate among residents, including long and unpredictable work shifts, sleep deprivation, burnout and impaired cognitive performance, high-stress situations (e.g., life and death emergencies), and easy access to means of self-harm [15,17,22].

The predictors of suicidal thoughts and behaviors among medical residents may vary across different regions, cultures, and contexts. Prior to the COVID-19 pandemic, specific data pertaining to suicide rates among Iranian medical residents was scarce. A comprehensive review and meta-analysis conducted in 2016 indicated a prevalence of 11.1% (95% CI, 9.0% to 13.7%) in suicidal ideation among medical students [23]. Recent years, however, have witnessed a

concerning escalation in suicide cases among medical residents. Reports from various Iranian news outlets have highlighted that between 2019 and June 2021, there were 15 documented suicide cases involving physicians and medical residents [24]. This trend is particularly alarming, with a significant number of these cases occurring in a relatively short timeframe, indicating an increase in such incidents compared to historical data. Contrary to the anticipated decrease in suicide rates among medical professionals following the subsidence of the COVID-19 pandemic, the reality has been a continued prevalence of these tragic events [25]. Contributing factors to this persistent issue include heightened workloads, substantial patient care duties, financial pressures, societal stigma surrounding mental health issues, and inadequate support structures within the healthcare system [24,26,27]. However, a comprehensive assessment of the dimensions of this issue is yet to be conducted. Hence, this study aimed to shed light on factors associated with suicidal ideation among medical residents in Iran. Understanding the unique challenges medical residents face in Iran is crucial for developing effective interventions and support systems to address their mental health needs.

## Methods

### Study design

This cross-sectional survey aimed to explore the factors associated with suicidal ideation among medical residents from different specialties in Tehran, the capital of Iran, during the COVID-19 pandemic. The survey used a combination of standardized scales and demographic questions to measure the variables of interest.

### Setting

The study was conducted online due to COVID-19 related social distancing measures, ensuring easy access for the majority of medical residents. The online questionnaire was distributed through WhatsApp groups to medical university residents in Tehran. It was available 24/7 from Jan. 1, 2022, to Feb. 28, 2022, and concluded upon reaching the desired sample size.

### Participants

The study included medical residents from various specialized fields across Tehran Medical Universities, namely Tehran University of Medical Sciences (TUMS), Shahid Beheshti University of Medical Sciences (SBUMS), Iran University of Medical Sciences (IUMS), AJA University of Medical Sciences and University of Social Welfare and Rehabilitation Sciences (USWR), representing different stages of residency training. According to recent information from the Iranian Ministry of Health, and Medical Education, these universities together accommodate around 4,000 medical residents, which accounts for about one-third of the nation's total number of medical residents [28]. Based on their field of practice, they were categorized into three groups: procedural specialties (including general surgery, neurosurgery, gynecology, emergency medicine, otolaryngology, urology, Orthopedics, and ophthalmology), clinical specialties (compromising internal medicine, psychiatry, neurology, pediatrics, radio-oncology, sports medicine, social medicine, and cardiology), and diagnostic specialties (encompassing radiology, pathology, and nuclear medicine).

The introductory page of the questionnaire explicitly outlined the study's purpose. The inclusion criteria were detailed on the second page, which required respondents to confirm their status as residents currently enrolled at the specified universities in Tehran and their consent to participate in the study. Only those who answered affirmatively to these initial queries were granted access to the full questionnaire, ensuring that the study focused on a specific and

relevant participant group. The exclusion criteria for this study comprised individuals who did not provide informed consent to participate in the research and those who were absent from the teaching hospital for three months preceding the study. In the absence of existing research, we used a conservative 50% prevalence rate for our sample size calculation, specifically referring to suicidal ideation among medical residents to encompass both low and high suicide risk levels. Employing a statistical power analysis with a 5% margin of error and a 95% confidence level, we determined that a sample size of 351 participants was optimal. A convenience sampling method was utilized in this study.

## Measures

The online questionnaire comprised a total of 64 items encompassing various domains, including demographic data, specialty, year of residency, suicidal ideation, having children, living arrangements, accommodation, history of psychiatric disorders, alcohol/substance use, family history of suicide attempts, personal history of suicide attempts, history of self-mutilation, number of shifts in a month, COVID-19 work exposure, death of close persons because of COVID-19, use of mental health services during COVID-19 pandemic, income, and components of the Beck Scale for Suicidal Ideation (BSSI), as well as items from the Depression, Anxiety, and Stress Scale-21 (DASS-21).

**Beck Scale for Suicidal Ideation (BSSI).**   The Beck Scale for Suicidal Ideation (BSSI) is a widely recognized and valuable tool for assessing suicidal ideation and behavior. It was developed in 1979 by Aaron T. Beck and consists of a 19-item self-report questionnaire that measures three dimensions of suicidal ideation: severity, frequency, and intent. Each item is rated on a scale of 0 to 2, indicating the absence, mild presence, or strong presence of specific suicidal thoughts. The total BSSI score ranges from 0 to 38 [29].

The first five items of the BSSI can be used to identify individuals with suicidal thoughts. If patients score zero on the Active Suicidal Desire item (#4) or the Passive Suicidal Desire item (#5), indicating no active or passive intention to die, they skip the remaining 14 items. Otherwise, the remaining 14 items are rated. Patients with active or passive ideation about killing themselves are considered suicide ideators [30].

There is no universally defined cut-off score for BSSI to determine suicide risk levels; however, studies have proposed various criteria based on empirical evidence and clinical judgment. In this regard, scores of 0 on all 19 BSSI items indicate no risk of suicide. Scores ranging from 0 to 5 indicate low risk, while scores of 6 and above indicate high risk [31].

Extensive research has substantiated the robust psychometric properties of the BSSI, demonstrating high internal consistency (Cronbach's alpha coefficients between 0.89 and 0.96) and satisfactory test-retest reliability (correlation coefficients above 0.80) [32]. In the study conducted by Esfahani et al., the Persian version of the Beck Scale for Suicidal Ideation (BSSI) utilized in this research exhibited substantial internal consistency, as indicated by Cronbach's alpha values ranging from 0.829 to 0.837. Furthermore, this version of the BSSI demonstrated significant correlations with depressive symptoms and revealed a unidimensional factor structure [33].

**Depression, Anxiety, and Stress Scale-21 (DASS-21).**   The DASS-21 is a screening instrument designed to assess symptoms of depression, anxiety, and stress experienced in the week leading up to the evaluation. It consists of three subscales (depression, anxiety, and stress), each comprising seven items. The final score for each subscale is calculated by summing the scores of the corresponding items, which are rated on a scale of 0 (does not apply to me at all) to 3 (applies to me very much or most of the time) [34]. The DASS-21 has demonstrated good internal consistency and reliability, as validated in various languages and populations [35].

The Persian DASS-21 study demonstrated high internal consistency (Cronbach's alpha: anxiety 0.79, stress 0.91, depression 0.93), robust test-retest reliability (0.740–0.881), and satisfactory model fit in confirmatory factor analysis (RMSEA 0.078, CFI 0.917), with ICCs from 0.75 to 0.86 confirming reliability in a nurse sample [36].

## Statistical analysis

The data were analyzed using SPSS software version 25. Descriptive statistics such as frequency and percentage were used to summarize the characteristics of the participants and their scores on the BSSI and the DASS-21. A Chi-square test was used to examine the association between suicidal ideation and other clinical and demographic variables. Multiple linear regression analysis was employed to determine the predictors of suicidal ideation, based on variables significantly correlated in the Chi-square test, and to adjust for potential confounders in our examination of suicidal ideation among medical residents. The level of significance was set at 0.05 for all tests. Cases with missing data were entirely excluded from the analysis.

## Ethical considerations

The research protocol received approval from the Ethics Committee of the University of Social Welfare and Rehabilitation Sciences (Code: IR.USWR.REC.1400.226) and was conducted in adherence to the principles outlined in the Declaration of Helsinki. The questionnaire clearly explained the study's objectives, and participants were required to provide online written informed consent. No personally identifiable information was collected, and authors did not have access to information that could identify individual participants during or after data collection. However, participants could provide their email addresses if they were interested in receiving study results. For individuals identified as being at moderate to high risk of suicide, we sent proactive messages to their respective email addresses, urging them to seek counseling with a volunteer psychiatrist coauthor who was available to assist.

## Results

Out of the total of 359 residents who responded to the survey, six residents did not complete it. As a result, the final number of participants included in the study was 353. The analysis included data from all these participants. The results show that most of the participants were female, aged 30–34 years, married, clinical specialists, in their first year of residency, without children, living with relatives, renting, or owning a home, without a history of psychiatric disorders, alcohol/substance use, family history of suicide attempts, personal history of suicide attempts or self-mutilation. The average number of shifts in a month was 0–7, and the level of COVID-19 work exposure was moderate. Most of the participants had a history of COVID-19 infection but were not admitted due to COVID-19 infection or the death of close persons because of COVID-19. About one-third of the participants used mental health services during the COVID-19 pandemic and had an income of less than 6 million Tomans per month (≈200 €).

The BSSI scores from our study revealed that the majority of participants did not show suicidal ideation, while a notable portion exhibited low and high-risk suicidal thoughts. Additionally, the prevalence of depression, anxiety, and stress was significant among participants, with a substantial number experiencing moderate to very severe levels of these conditions. Further demographic and clinical characteristics can be found in Table 1.

The association between suicidal ideation and other clinical and demographic variables was examined using Pearson Chi-Square test. Table 2 presents the results of the analysis.

The analysis showed that factors such as gender, marital status, type of specialty, residency year, having children, living arrangements, type of accommodation, history of COVID-19

**Table 1. Demographic and clinical characteristics of participants.**

| Characteristic | | Frequency | Percent |
|---|---|---|---|
| **Gender** | Female | 268 | 75.9 |
| | Male | 85 | 24.1 |
| **Age** | 25–29 | 125 | 35.4 |
| | 30–34 | 177 | 50.1 |
| | 35–39 | 33 | 9.3 |
| | 40 years and above | 18 | 5.1 |
| **Marital status** | Married | 205 | 58.1 |
| | Single | 132 | 37.4 |
| | Divorced | 16 | 4.5 |
| **Specialty** | Procedural | 84 | 23.8 |
| | Clinical | 239 | 67.7 |
| | Diagnostic | 30 | 8.4 |
| **Year of residency** | 1 | 111 | 31.4 |
| | 2 | 83 | 23.5 |
| | 3 | 84 | 23.8 |
| | 4 and above | 75 | 21.2 |
| **Having children** | No | 298 | 84.4 |
| | Yes | 55 | 15.6 |
| **Living arrangement** | Spouse | 71 | 20.1 |
| | Relatives | 247 | 70.0 |
| | Peers | 35 | 9.9 |
| **Accommodation** | Dormitory | 32 | 9.1 |
| | Rental home | 157 | 44.5 |
| | Personal home | 164 | 46.5 |
| **History of psychiatric disorders** | No | 224 | 63.5 |
| | Yes | 129 | 36.5 |
| **History of alcohol/substance use** | No | 273 | 77.3 |
| | Yes | 80 | 22.7 |
| **Family history of suicide attempts** | No | 302 | 85.6 |
| | Yes | 51 | 14.4 |
| **Personal history of suicide attempts** | No | 334 | 94.6 |
| | Yes | 19 | 5.4 |
| **History of self- mutilation** | No | 235 | 94.9 |
| | Yes | 18 | 5.1 |
| **Number of shifts in a month** | 0–7 | 175 | 49.6 |
| | 8–14 | 161 | 45.6 |
| | 15 and above | 17 | 4.8 |
| **COVID-19 Work Exposure** | High exposure (Working in COVID-19 wards or respiratory emergency) | 89 | 25.2 |
| | Moderate exposure (Occasional visits or consultations in the COVID-19 ward) | 152 | 43.1 |
| | No exposure | 112 | 31.7 |
| **History of COVID-19 infection** | No | 92 | 26 |
| | Yes | 261 | 74 |
| **History of admission due to COVID-19 infection** | No | 336 | 95.2 |
| | Yes | 17 | 4.8 |
| **Death of Close Persons because of COVID-19** | No | 201 | 56.9 |
| | Yes | 152 | 43.1 |

*(Continued)*

**Table 1.** (Continued)

| Characteristic | | Frequency | Percent |
|---|---|---|---|
| **Use of Mental Health Services during COVID-19 Pandemic** | No | 246 | 69.7 |
| | Yes | 107 | 33 |
| **Income per month** | Less than 6 million Tomans (≈200 €) | 226 | 64 |
| | 6–9 million Tomans (≈ 200–300 €) | 101 | 28.6 |
| | More than 9 million Tomans (≈300 €) | 26 | 7.4 |
| **BSSI** | No suicidal ideation | 232 | 65.7 |
| | Low risk suicidal ideation | 85 | 24.1 |
| | High risk suicidal ideation | 36 | 10.2 |
| | Overall prevalence | 121 | 34.3 |
| **Depression** | No depression | 108 | 30.6 |
| | Mild | 45 | 12.7 |
| | Moderate | 110 | 31.2 |
| | Severe | 33 | 9.3 |
| | Very severe | 57 | 16.1 |
| **Anxiety** | No anxiety | 130 | 36.8 |
| | Mild | 42 | 11.9 |
| | Moderate | 112 | 31.7 |
| | Severe | 31 | 8.8 |
| | Very severe | 38 | 10.8 |
| **Stress** | No stress | 125 | 35.4 |
| | Mild | 68 | 19.3 |
| | Moderate | 77 | 21.8 |
| | Severe | 65 | 18.4 |
| | Very severe | 18 | 5.1 |

Note: Percentages are based on the total number of participants (N = 353). BSSI: Beck Scale for Suicide Ideation.

infection, and history of hospitalization due to COVID-19 were not statistically significant in their association with suicidal ideation.

History of psychiatric disorders, alcohol/substance use, family history of suicide attempts, personal history of suicide attempts, history of self-mutilation, and number of shifts in a month were all significantly associated with suicidal thoughts. COVID-19 work exposure was also significantly associated with suicidal ideation, with high exposure being more prevalent among those with high-risk suicidal ideation than those with no or low risk suicidal ideation.

Death of close persons because of COVID-19 and use of mental health services during the COVID-19 pandemic were significantly associated with suicidal ideation, with higher use among those with high risk than those with no or low risk suicidal ideation.

Income was marginally significantly associated with suicidal ideation, with lower income being more prevalent among those with high risk than those with no or low risk suicidal ideation.

Depression, anxiety, and stress were all significantly associated with suicidal ideation, with higher levels of depression, anxiety, and stress being more prevalent among those with high risk than those with no or low risk suicidal ideation.

A linear regression analysis was conducted to identify the predictors of suicide among the variables studied. Based on the Chi-square test results, only the statistically significant variables, and the continues score of suicide ideation were included in the regression model to examine the amount of variance explained (Table 3).

**Table 2. Association between suicidal ideation and other clinical and demographic variables.**

| Variables | | No suicidal ideation | Low risk suicidal ideation | High risk suicidal ideation | Pearson Chi-Square Value | df | P value |
|---|---|---|---|---|---|---|---|
| **Gender** | Female | 173(64.6%) | 69(25.7%) | 26(9.7%) | 1.786 | 2 | .410 |
| | male | 59(64.1%) | 16(18.8%) | 10(11.8%) | | | |
| **Age** | 25–29 | 86(68.8%) | 26(20.8%) | 13(10.4%) | 7.321 | 4 | .560 |
| | 30–34 | 115(65%) | 46(26%) | 16(9%) | | | |
| | 35–39 | 19(57.6%) | 9(27.3%) | 5(15.1%) | | | |
| | 40 years and above | 12(66.7%) | 4((22.2%) | 2(11.1%) | | | |
| **Marital status** | Married | 142(69.3%) | 45(22%) | 18(8.8%) | 7.149 | 4 | .128 |
| | Single | 84(63.7%) | 33(25%) | 15(11.4%) | | | |
| | Divorced | 6(37.5%) | 7(43.8%) | 3(18.7%) | | | |
| **Specialty** | Procedural | 46(54.8%) | 28(33.3%) | 10(11.9%) | 6.658 | 4 | .155 |
| | Clinical | 164(68.7%) | 51(21.3%) | 24(10%) | | | |
| | Diagnostic | 22(73.3%) | 6(20%) | 2(6.7%) | | | |
| **Year of residency** | 1 | 77(69.4%) | 25(22.5%) | 9(8.1%) | 5.081 | 6 | .533 |
| | 2 | 50(60.2%) | 24(29%) | 9(10.8%) | | | |
| | 3 | 51(60.7%) | 21(25%) | 12(14.3%) | | | |
| | 4 and above | 54(72%) | 15(20%) | 6(8%) | | | |
| **Having children** | No | 196(65.8%) | 71(23.8%) | 31(10.4%) | .130 | 2 | .937 |
| | Yes | 36(65.4%) | 14(25.5%) | 5(9.1%) | | | |
| **Living arrangement** | Spouse | 44(62%) | 17(24%) | 10(14%) | 2.298 | 4 | .681 |
| | Relatives | 167(67.6%) | 58(23.5%) | 22(8.9%) | | | |
| | Peers | 21(60%) | 10(28.6%) | 4(11.4%) | | | |
| **Accommodation** | Dormitory | 21(65.6%) | 7(21.9%) | 4(12.5%) | 3.116 | 4 | .539 |
| | Rental home | 98(62.5%) | 39(24.8%) | 20(12.7%) | | | |
| | Personal home | 113(68.9%) | 39(23.8%) | 12(7.31%) | | | |
| **History of psychiatric disorders** | No | 161(45%) | 48(15%) | 15(4%) | 12.690 | 2 | **.002** |
| | Yes | 71(20%) | 37(10%) | 21(6%) | | | |
| **History of Alcohol/substance use** | No | 191(54%) | 55(16%) | 27(8%) | 11.146 | 2 | **.004** |
| | Yes | 41(12%) | 30(8%) | 9(2%) | | | |
| **Family history of suicide attempts** | No | 209(59%) | 65(19%) | 28(8%) | 11.291 | 2 | **.004** |
| **Personal history of suicide attempts** | Yes | 23(6%) | 20(5%) | 8(3%) | | | |
| | No | 226(64%) | 81(22%) | 27(8%) | 30.843 | 2 | **.001** |
| | Yes | 6(2%) | 4(1%) | 9(3%) | | | |
| **History of self- mutilation** | No | 228(64%) | 77(22%) | 30(9%) | 18.682 | 2 | **.001** |
| | Yes | 4(1%) | 8(3%) | 6(2%) | | | |
| **Number of shifts in a month** | 0–7 | 126(72%) | 35(20%) | 14(8%) | 8.941 | 4 | **.049** |
| | 8–14 | 99(61.5%) | 43(26.7%) | 19(11.8%) | | | |
| | 15 and above | 7(41.2%) | 7(41.2%) | 3(17.6%) | | | |
| **COVID-19 Work Exposure** | Moderate exposure (Occasional visits or consultations in the COVID-19 ward) | 109(31%) | 29(8%) | 14(4%) | 13.085 | 4 | **.011** |
| | High exposure (Working in COVID-19 wards or respiratory emergency) | 45(13%) | 30(8%) | 14(4%) | | | |
| | No exposure | 78(22%) | 26(7%) | 8(3%) | | | |
| **History of COVID-19 infection** | No | 61(17%) | 23(6%) | 8(3%) | .326 | 2 | .850 |
| | Yes | 171(48%) | 62(18%) | 28(8%) | | | |

*(Continued)*

**Table 2.** (Continued)

| Variables | | No suicidal ideation | Low risk suicidal ideation | High risk suicidal ideation | Pearson Chi-Square Value | df | *P* value |
|---|---|---|---|---|---|---|---|
| **History of admission due to COVID-19 infection** | No | 223(63%) | 80(22%) | 33(9%) | 1.627 | 2 | .443 |
| | Yes | 9(3%) | 5(2%) | 3(1%) | | | |
| **Death of Close Persons because of COVID-19** | No | 153(43%) | 37(10%) | 11(3%) | 24.13 | 2 | **.001** |
| | Yes | 79(23%) | 48(14%) | 25(7%) | | | |
| **Use of Mental Health Services during COVID-19 Pandemic** | No | 166(47%) | 63(18%) | 17(5%) | 9.77 | 2 | **.008** |
| | Yes | 66(19%) | 22(6%) | 19(5%) | | | |
| **Income per month** | Less than 6 million Tomans (≈200 €) | 141(40%) | 57(16%) | 28(8%) | 9.200 | 4 | **.049** |
| | 6–9 million Tomans (≈200–300 €) | 74(22%) | 19(5%) | 4(1%) | | | |
| | More than 9 million Tomans(≈300 €) | 19(5%) | 7(2%) | 2(1%) | | | |
| **Depression** | No depression | 102(29%) | 6(2%) | 0 | 146.128 | 8 | **.001** |
| | Mild | 37(10%) | 7(2%) | 1(1%) | | | |
| | Moderate | 63(18%) | 43(13%) | 4(1%) | | | |
| | Severe | 14(4%) | 13(4%) | 6(2%) | | | |
| | Very severe | 16(4%) | 16(4%) | 25(6%) | | | |
| **Anxiety** | No anxiety | 107(30%) | 21(6%) | 2(0.5%) | 81.342 | 8 | **.001** |
| | Mild | 29(8%) | 11(3%) | 2(0.5%) | | | |
| | Moderate | 61(17%) | 37(10%) | 14(4%) | | | |
| | Severe | 22(6%) | 4(1%) | 5(2%) | | | |
| | Very severe | 13(4%) | 12(4%) | 13(4%) | | | |
| **Stress** | No stress | 105(30%) | 18(5%) | 2(0.5%) | 81.342 | 8 | **.001** |
| | Mild | 46(14%) | 20(6%) | 2(0.5%) | | | |
| | Moderate | 49(14%) | 21(6%) | 7(2%) | | | |
| | Severe | 25(7%) | 24(7%) | 16(4%) | | | |
| | Very severe | 7(2%) | 2(0.5%) | 9(3%) | | | |

Significant values are in bold (P<0.05).

The analysis revealed that suicide ideation was significantly predicted by a history of alcohol/substance use, family history of suicide attempts, personal history of suicide attempts, history of self-mutilation, number of shifts in a month, death of close persons because of COVID-19, income, and depression (all p-values < 0.05). All these variables, except for income, had positive associations with suicide ideation, indicating that higher levels or categories of these variables corresponded to higher levels of suicide ideation. The most influential predictor was depression (Beta = 0.604), followed by COVID-19 work exposure (Beta = 0.203) and a history of self-mutilation (Beta = 0.191).

## Discussion

This study examined the factors associated with suicidal ideation among medical residents in Iran during the third and fourth waves of the COVID-19 pandemic. The findings revealed that a significant proportion of participants (34.3%) reported having suicidal ideas, with 10.2% indicating high-risk suicidal thoughts. These rates were higher than those reported in the general population of Iran (12.7–14%) and in medical students (7–26%) in previous studies [37–40]. The prevalence of suicidal thoughts among medical residents in this study also surpassed the rates reported in recent studies conducted in other countries [10,41,42].

**Table 3. Regression coefficients of suicide ideation\* and dependent variables.**

| Model | Unstandardized Coefficients | | Standardized Coefficients | t | p value |
|---|---|---|---|---|---|
| | B | Std. Error | Beta | | |
| (Constant) | -.246 | .482 | | -.510 | .611 |
| History of psychiatric disorders | -1.007 | 2.462 | .031 | .409 | .683 |
| History of Alcohol/substance use | .197 | .094 | .177 | 2.098 | **.038** |
| Family history of suicide attempts | 3.149 | 1.187 | .136 | 2.654 | **.008** |
| Personal history of suicide attempts | 4.161 | 2.073 | .147 | 2.007 | **.047** |
| History of self- mutilation | 7.097 | 1.926 | .191 | 3.685 | **.001** |
| Number of shifts in a month | .230 | .99 | .127 | 2.325 | **.021** |
| COVID-19 Work Exposure | 1.930 | 1.081 | 0.203 | 1.176 | 0.75 |
| Death of Close Persons because of COVID-19 | 1.916 | .835 | .117 | 2.295 | **0.02** |
| Use of Mental Health Services during COVID-19 Pandemic | 1.360 | .928 | 0.75 | 1.466 | 0.144 |
| Income | -.155 | .071 | -.175 | -2.185 | **.031** |
| Depression | .561 | .113 | .604 | 4.964 | **.001** |
| Anxiety | .144 | .126 | .120 | 1.150 | .253 |
| Stress | . 062 | .153 | .053 | -.402 | .688 |

Significant values are in bold (P<0.05)

\*The continuous score of suicide ideation was used in analyses.

The study further explored the levels of depression, anxiety, and stress experienced by the participants. The findings revealed that more than 50% of the medical residents reported moderate to severe symptoms. This aligns with the results of a recent study conducted on 150 medical residents from Tehran University of Medical Sciences in Iran, where 23% of residents reported severe to highly severe depression, 24.9% reported severe to highly severe anxiety, and 33.8% reported severe to highly severe stress [6]. Notably, these percentages significantly surpass those reported in pre-pandemic studies. For instance, a study conducted in 2007 reported depression in only 15.6% of medical residents in Tehran [43], while another study in 2010 found that 92% of Iranian medical residents did not experience anxiety symptoms [44]. Consequently, the high prevalence of depression and anxiety observed in our study raises concerns. Several factors contribute to this phenomenon. The COVID-19 pandemic is a significant factor, as it has posed unprecedented challenges and stressors for healthcare workers, especially those in training [45]. The socio-economic challenges faced by Iran in recent years may have also impacted the well-being of healthcare workers [46]. Moreover, mental health stigma in the medical community discourages residents from seeking help and discussing their struggles openly. The fear of being perceived as weak hinders them from seeking professional support [47]. Additionally, the demanding nature of their training and long working hours limit their ability to prioritize mental well-being or access mental health services [8].

Among the demographic and personal factors assessed, a history of psychiatric disorders, alcohol/substance use, family history of suicide attempts, personal history of suicide attempts, and history of self-mutilation were positively related to suicidal ideation. These factors indicate higher vulnerability and lower resilience in coping with stress and have been previously identified as risk factors for suicidal behavior across various populations [7,8,10,15]. Medical residents with such characteristics require enhanced attention and support from mental health professionals and institutions [20,48]. Contrary to previous studies conducted in the general population of Iran [37,38,49], demographic factors such as gender, marital status, number of children,

living arrangement, and accommodation did not show a significant relationship with suicidal ideation in our study. This suggests potential differences in the risk factors for suicidal ideation between the general population and medical residents in Iran. Our study did not identify a significant variation in the frequency of suicidal thoughts among different age groups of the participants. However, it is essential to acknowledge that the age range of the participating residents was relatively narrow and that this age range has been consistently identified as having the highest susceptibility to suicide in prior studies conducted among the general population [38,49,50].

Regarding work-related factors, our study found a positive correlation between the number of shifts per month, COVID-19 work exposure, and suicidal ideation among medical residents. This observation necessitates a comparison of working hours for medical residents in Iran with global standards. Notably, Iranian medical residents often have longer working hours compared to their counterparts in countries like the US, Canada, and Turkey [51]. The ACGME, for instance, limits resident work hours to 80 per week, restricts continuous work to 24 hours, and mandates significant rest periods between shifts and days off. In contrast, Iranian residents frequently work beyond these limits, reflecting the intense workload and stress they face, which is further exacerbated during pandemic conditions [51,52]. This situation, coupled with high responsibility and exposure to traumatic events, has been linked to increased mental health problems and suicidal ideation in healthcare workers [45,53]. Our study's lack of significant variation in suicidal ideation across residency years contrasts with other research showing notable differences. For instance, a study on ACGME-Accredited Programs' residents found higher suicide rates early in residency, especially during specific academic months [19]. This parallels a study of 740 interns, where a 370% increase in suicidal ideation was noted in the initial three months of internship, highlighting the stress associated with transitions in medical training [54]. Several factors specific to our study context and methodology could explain this discrepancy. Firstly, the uniformly high stress levels across all residency years in our study, possibly exacerbated by the COVID-19 pandemic, may have obscured any year-specific variations. Additionally, cultural factors in Iran, such as potential underreporting due to mental health stigma, could have influenced residents' reporting of mental health issues, masking any differences in suicidal ideation [55]. Lastly, the composition and size of our sample, particularly the subgroups for each residency year, might not have been sufficient to detect significant differences. Although our study did not find a significant difference in suicide risk across various medical specialties, earlier research has indicated a heightened risk in specific fields such as anesthesiology, general surgery, and internal medicine [53]. This lack of disparity in our findings could potentially be related to the size and specialty distribution of our sample. The representation of different specialties in our study might not have been sufficiently comprehensive to detect nuanced variations in suicide risk.

The findings of this study also indicated a positive association between suicidal thoughts and the loss of close persons because of COVID-19. This factor indicates the personal loss and grief some medical residents may experience during the pandemic, affecting their mental health and well-being. Bereavement resulting from COVID-19 has been associated with a heightened risk of depression, anxiety, post-traumatic stress disorder, and complicated grief [56].

Among the economic factors, income was negatively related to suicidal ideation. This factor reflects the financial difficulties that some medical residents may face, which may add to their stress [52].

Depression emerged as the primary predictor of suicidal ideation among medical residents, highlighting the profound emotional distress that some individuals in this group may endure, leading to impaired functioning and diminished quality of life. Extensive evidence has consistently shown that psychiatric disorders, particularly depression, substantially elevate the risk of suicide across diverse populations, including healthcare workers [37,42,57].

However, many training programs have faced challenges in identifying and offering treatment to medical students and residents at risk [58]. These findings underscore the significance of screening for depression and delivering effective treatment to medical residents experiencing depression.

The study has several limitations that should be acknowledged. First, the study used a cross-sectional design, which limits the causal inference between the variables. Longitudinal studies are needed to examine the temporal relationship between suicidal thoughts and other factors among medical residents. Second, the study relied on self-report measures, which may be subject to recall and social desirability biases. Objective measures such as clinical interviews are needed to validate the self-report measures. Third, the use of convenience sampling in our study may not have adequately represented all medical specialties, limiting comparative analysis. Furthermore, the focus on medical residents from a single city narrows the generalizability of our results. Representative samples from various regions or cities are necessary to understand cross-cultural variations in suicidal thoughts among medical residents. The fourth limitation of our study is the omission of additional risk factors that could potentially influence the outcomes as confounding factors. For instance, we did not assess the impact of personality traits, marital discord, conflicts with family or colleagues, sleep quality, chronic medical conditions, and the potential protective role of social support and religious beliefs. These factors have been identified in previous research as potential contributors to mental health outcomes [59–61] and could have provided a more comprehensive understanding of the associations observed in our study. Additionally, the study did not verify respondent authenticity, introducing potential uncertainty in the findings and emphasizing the need for future research to confirm the reliability of data. In the realm of future research, the inclusion of institutional reviews conducted in collaboration with medical students' union councils and medical specialty residents' associations within individual universities holds great promise. These reviews can serve as invaluable tools for gaining a comprehensive understanding of the multifaceted landscape encompassing the needs, challenges, and overall satisfaction levels of medical residents.

## Conclusions

This study casts vital light on the urgent mental health issues among Iranian medical residents during the COVID-19 pandemic, providing new insights into the prevalence and drivers of suicidal ideation in a previously underexplored context. The high rates of suicidal thoughts found in this study highlight the urgent need for targeted mental health interventions, particularly focusing on depression as a key predictor. Addressing these challenges through effective depression screening and treatment is crucial for both improving the mental health and sustaining the professional effectiveness of these vital healthcare workers.

## Supporting information

**S1 Checklist. STROBE statement—Checklist of items that should be included in reports of observational studies.**
(DOCX)

## Acknowledgments

We would like to express our sincere gratitude to all the participants who took part in this cross-sectional survey.

## Author Contributions

**Conceptualization:** Fahimeh Saeed, Mohammadreza Shalbafan.

**Data curation:** Fahimeh Saeed, Elaheh Ghalehnovi, Mahdieh Saeidi.

**Formal analysis:** Neda Ali beigi, Mohsen Vahedi.

**Funding acquisition:** Fahimeh Saeed.

**Investigation:** Elaheh Ghalehnovi, Mahdieh Saeidi.

**Methodology:** Fahimeh Saeed, Elaheh Ghalehnovi, Neda Ali beigi, Mohsen Vahedi, Mohammadreza Shalbafan, Sheikh Shoib.

**Project administration:** Fahimeh Saeed.

**Software:** Mohsen Vahedi.

**Supervision:** Fahimeh Saeed, Sheikh Shoib.

**Validation:** Leila Kamalzadeh, Ali Nazeri Astaneh, Amir Hossein Jalali Nadoushan.

**Visualization:** Ali Nazeri Astaneh, Amir Hossein Jalali Nadoushan, Sheikh Shoib.

**Writing – original draft:** Leila Kamalzadeh.

**Writing – review & editing:** Leila Kamalzadeh, Sheikh Shoib.

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
