## [Decision Letter · Decision Letter 0]

12 Dec 2023

PONE-D-23-19377Factors associated with suicidal ideation among medical residents in Iran during the COVID-19 pandemic: a multicentric cross-sectional surveyPLOS ONE

Dear Dr. Kamalzadeh,

Thank you for submitting your manuscript to PLOS ONE. After careful consideration, we feel that it has merit but does not fully meet PLOS ONE’s publication criteria as it currently stands. Therefore, we invite you to submit a revised version of the manuscript that addresses the points raised during the review process. Please submit your revised manuscript by Jan 26 2024 11:59PM. If you will need more time than this to complete your revisions, please reply to this message or contact the journal office at plosone@plos.org. Please include the following items when submitting your revised manuscript:A rebuttal letter that responds to each point raised by the academic editor and reviewer(s). You should upload this letter as a separate file labeled 'Response to Reviewers'.A marked-up copy of your manuscript that highlights changes made to the original version. You should upload this as a separate file labeled 'Revised Manuscript with Track Changes'.An unmarked version of your revised paper without tracked changes. You should upload this as a separate file labeled 'Manuscript'.

We look forward to receiving your revised manuscript.

Kind regards,

Juan Jesús García-Iglesias, Ph.D.

Academic Editor

PLOS ONE

Journal Requirements:

**Additional Editor Comments:**

Please revise the manuscript according to the reviewers' comments and upload the revised file. Any revisions should be “clearly highlighted”, for example using the Track Changes function in Microsoft Word or using a different colour, so that they are easily visible to the editors and reviewers.

Please provide a short Cover letter detailing any changes, for the benefit of the editors and reviewers.

One of our associate editors and two reviewers carefully read the manuscript. Based on their evaluations the manuscript is major revision. The associate editor provided the following reasons:

The manuscript needs to be rewritten, taking into account the following main comments made by the reviewers

Introduction: restructuring of the text to provide more coherent and connected ideas and sections, including relevant references for this topic.

Methods: more details are needed.

Results: synthesize findings and present them in a systematic way.

Discussion: do not discuss main factors, secondary factors, generalizability, recommendations and implications

Reviewers' comments:

Reviewer's Responses to Questions

**Comments to the Author**

1. Is the manuscript technically sound, and do the data support the conclusions?

Reviewer #1: Partly

Reviewer #2: Partly

2. Has the statistical analysis been performed appropriately and rigorously? 

Reviewer #1: Yes

Reviewer #2: No

3. Have the authors made all data underlying the findings in their manuscript fully available?

Reviewer #1: Yes

Reviewer #2: No

4. Is the manuscript presented in an intelligible fashion and written in standard English?

Reviewer #1: No

Reviewer #2: Yes

5. Review Comments to the Author

Reviewer #1: The manuscript titled ' Factors associated with suicidal ideation among medical residents in Iran during the COVID-19 pandemic: a multicentric cross-sectional survey' investigated the factors associated with suicidal ideation among medical residents in Iran during the

COVID-19 pandemic.

This is an interesting subject for research because suicidal ideation in this group can vary over time and may be influenced by various factors, including societal, cultural, and healthcare system-related factors.

Title:

The authors conducted a cross-sectional online survey among medical residents in various specialities in Tehran. There are three Medical universities in Tehran which accommodate less than a third of all medical residents in Iran. The title should be modified based on the actual setting of this study that has been mentioned in the Methods section.

https://arakmu.ac.ir/vcsc/fa/news/27424/%D8%B8%D8%B1%D9%81%DB%8C%D8%AA-%DB%B4-%D9%87%D8%B2%D8%A7%D8%B1-%D9%86%D9%81%D8%B1%DB%8C-%D8%A8%D8%B1%D8%A7%DB%8C-%D9%BE%D8%B0%DB%8C%D8%B1%D8%B4-%D8%AF%D8%B3%D8%AA%DB%8C%D8%A7%D8%B1-%D8%AA%D8%AE%D8%B5%D8%B5%DB%8C-%D9%BE%D8%B2%D8%B4%DA%A9%DB%8C

Introduction:

1- Consider including the reporting of 95% confidence intervals (CI) for a meta-analysis's 1- overall prevalence to provide readers with more informative data.

2- A reader expects to hear about the history of suicides in this population of doctors in Iran before and after the Covid-19 pandemic, such as the information in reference 24 (http://dx.doi.org/10.5812/ijhrba.117651).

Methods:

1- The authors could consider providing a more precise period of study instead of using the term "amidst" the COVID-19 pandemic. Include relevant dates for the online survey, recruitment periods, online access, and closing dates.

2- Please elaborate on the methods used to ensure the authenticity of participants in the online survey, such as email verification, access control, confirmation questions, etc. Also, provide the eligibility criteria and sources of participant selection.

3- Please clarify the measure domains to which the 50% prevalence rate used to calculate the sample size belongs.

4- Please include the psychometrics measures used for the Persian version of the tools used in the study.

5- Are there potential confounders for this study?

6- Please provide diagnostic criteria/threshold, such as low-high and no suicidal ideation.

7- Please describe efforts to address potential sources of bias.

8- Describe those statistical methods used to control for confounding.

9- Were there any missing data? Please explain how missing data were addressed.

Results:

1- Please provide numbers of individuals at each stage of the study (e.g., eligible, included, completed the survey, analysed). Consider using a flow diagram if possible.

2- Please refrain from the redundant text in the results section that overlaps with the information presented in Table 1. The text should complement the table, not repeat its numbers.

Discussion:

1- Please address the confidence level regarding the relationship between higher reported anxiety or depression and the COVID-19 pandemic vs the country’s socio-economic factors.

2- Discussion: Discuss the potential impact of cultural issues and relationships among different levels of medical residents (Y1, Y2, Y3, …) on the findings.

3- A comparison of the number of working hours scheduled for medical residents in Iran to international standards could be beneficial to this discussion.

4- Address the external validity of the sampling method's results for medical speciality trainees in Tehran universities.

5- Please discuss the statistical power for some of the mentioned results, such as the non-significant disparity in the risk of suicide among different residency fields, etc.

6- Some parts of the discussion are out of the scope of the results of this study. Such as “COVID-19-related factors also affected the prevalence of suicidal thoughts among medical residents”. What is the role of other factors? Such as socio-economic status, timely access to most efficient Covid-19 vaccines etc.

One step in measuring suicidal ideation among doctors is conducting institutional reviews. With the three prominent medical universities with residency programmes in Tehran, an institutional review should be done to have a more comprehensive picture of this subject. There are Medical Students' Union Councils and even some associations of medical speciality residents in each university, and the authors should get their representatives' opinions in this regard for having a comprehensive study. It is also beneficial to suggest these for future research.

Reviewer #2: This is an important area of study. Medical workers rarely participate in research as participants yet they should benefit from research outputs. There is need for targeted interventions for health workers including the residents and the nursing or medical students.

1. What was the target population? How many medical residents did you sample from?

2. Under study design, you state that the data collection period was from 1 April to 10 March 2022, between the third and fourth waves of the COVID-19 pandemic in Iran. This is not clear. Was it April of 2022 or 2021?

3. The conclusions do not seem to be connected to the study findings. Please rewrite them.

4. Explain why you chose to use linear regression for analysis for categorical variables.

5. In results, you state that sex, marital status, specialty, year of residency, having children, living arrangement, accommodation, history of COVID-19 infection, and history of admission due to COVID-19 infection were not significantly associated with suicidal ideation (P > .05 for all variables). Please specify that you are referring to statistical significancy.

6. You collected data on death of a relative due to COVID-19. Did you also get information on death of a relative due to other causes?

7. Were there any statistically significant differences across specialties? You mentioned many but they are not reflected in the results.

8. What has this study contributed to the already existing literature. Please state/describe it.

9. Were there any differences across the different schools/universities? Please present this.

10. Please reformat your tables to fit on one page each.

6. PLOS authors have the option to publish the peer review history of their article (what does this mean?). If published, this will include your full peer review and any attached files.

Reviewer #1: No

Reviewer #2: No

---

## [Author Response · Author response to Decision Letter 0]

15 Jan 2024

Reviewer #1

Reviewers’ Comments Authors’ Response

Title The authors conducted a cross-sectional online survey among medical residents in various specialties in Tehran. There are three medical universities in Tehran which accommodate less than a third of all medical residents in Iran. The title should be modified based on the actual setting of this study that has been mentioned in the Methods section. 

Thank you for your insightful feedback. We have revised the title and corrected an oversight in our text where AJA University of Medical Sciences was inadvertently omitted, ensuring the inclusion of residents from all five medical universities in Tehran in our study.

Please refer to Lines 1 & 3 on Page 1.

Introduction

 1. Consider including the reporting of 95% confidence intervals (CI) for a meta-analysis's 1- overall prevalence to provide readers with more informative data. 

Per your recommendation, we added the 95% confidence intervals to the text.

Please refer to Lines 65 & 66 on Page 5.

 2. A reader expects to hear about the history of suicides in this population of doctors in Iran before and after the Covid-19 pandemic, such as the information in reference 24 (http://dx.doi.org/10.5812/ijhrba.117651). 

Following your insightful recommendation, we have updated the introduction in our paper to include detailed information about the trends in suicides among Iranian medical residents, both before and after the onset of the COVID-19 pandemic.

Please refer to Lines 81-94 on Page 6.

Methods 

1- The authors could consider providing a more precise period of study instead of using the term "amidst" the COVID-19 pandemic. Include relevant dates for the online survey, recruitment periods, online access, and closing dates. 

We apologize for the ambiguity in the text. The study's time frame has been corrected to indicate that the online survey was available 24/7 from Jan 1, 2022, to Feb 28, 2022, and concluded upon reaching the desired sample size.

Please refer to Lines 108 & 109 on Page 7.

 2- Please elaborate on the methods used to ensure the authenticity of participants in the online survey, such as email verification, access control, confirmation questions, etc. Also, provide the eligibility criteria and sources of participant selection. 

In our study on the sensitive topic of suicide and mental health among medical residents, we prioritized ethical considerations by forgoing stringent authenticity verification and access control measures like email checks or university record cross-referencing. This approach was adopted to ensure participant anonymity, minimize any potential distress, and encourage open and honest participation.

Per your suggestion, additional details on eligibility criteria and participant selection have been included in the methods section.

Please refer to Lines 124-128 on Page 8.

 3- Please clarify the measure domains to which the 50% prevalence rate used to calculate the sample size belongs. 

Due to the lack of existing research in our study area, a conservative prevalence rate of 50% was chosen for sample size determination. 

Please refer to Lines 131-133 on Page 8.

 4- Please include the psychometrics measures used for the Persian version of the tools used in the study. As recommended, we have now included the psychometric properties of the Persian versions of the assessment tools in the methods section.

Please refer to Lines 161-165, and 173-176 on Pages 9 & 10.

 5- Are there potential confounders for this study? 

In response to your inquiry about potential confounders in our study, we have outlined several limitations in the manuscript which could be considered as potential confounding factors. These limitations include cross-sectional design, reliance on self-report measures, sampling and representativeness, omission of additional risk factors.

Please refer to Lines 337-352 on Page 22.

 6- Please provide diagnostic criteria/threshold, such as low-high and no suicidal ideation. 

Suicide risk is defined by the BSSI score, which ranges from 0 to 38. A score of 0 on all 19 items denotes no risk, 0-5 denotes low risk, and 6 or higher denotes high risk.

Please refer to Lines 155-158 on Page 9.

 7-Please describe efforts to address potential sources of bias. 

To address potential biases in our study, we have taken several measures. Our sample, limited to Tehran, may not represent all regions, so we've emphasized the context-specific nature of our findings (please refer to study limitations on Page 22 – Lines 342-346). We used self-report measures, which might introduce recall and social desirability biases, and acknowledged this in our manuscript (please refer to study limitations on Page 22 - Lines 340-342). Our convenience sampling might not capture all medical specialties, potentially leading to selection bias (please refer to study limitations on Page 22 - Lines 342-346). We conducted rigorous statistical analysis to control for confounding variables and maintained ethical considerations for participant confidentiality and anonymity. These steps aimed to minimize bias and enhance the credibility of our findings.

 8- Describe those statistical methods used to control for confounding 

To mitigate potential confounding variables in our analysis, we employed multiple linear regression analysis, adjusting various independent variables simultaneously.

Please refer to Lines 181-184 on Page 10.

 9- Were there any missing data? Please explain how missing data were addressed. 

As mentioned in the Statistical Analysis section, cases with missing data were entirely excluded from the analysis.

Please refer to Line 185 on Page 10.

Results 

1- Please provide numbers of individuals at each stage of the study (e.g., eligible, included, completed the survey, analysed). Consider using a flow diagram if possible 

Our study's structure comprised a straightforward process, which did not necessitate a flow diagram. However, to provide a clear understanding of participant involvement, here are the specifics: Of the 359 medical residents who started the survey, six did not complete it, resulting in a final participant count of 353. The analysis included data from all these participants.

Please refer to Lines 198-200 on Page 11.

 2- Please refrain from the redundant text in the results section that overlaps with the information presented in Table 1. The text should complement the table, not repeat its numbers. 

In accordance with your feedback, we have removed the redundant numerical data from the results section.

Discussion

 1- Please address the confidence level regarding the relationship between higher reported anxiety or depression and the COVID-19 pandemic vs the country’s socio-economic factors. 

Acknowledging your feedback, our study did observe increased mental health issues among medical residents during the pandemic, yet it's unclear whether this is due to the pandemic itself or Iran's socio-economic challenges, as the study didn't separately analyze these factors with distinct confidence levels. Future research should include socio-economic considerations to thoroughly understand these influences. Your suggestion is highly valued for guiding our future research direction.

 2-Discuss the potential impact of cultural issues and relationships among different levels of medical residents (Y1, Y2, Y3, …) on the findings. 

Thank you for your valuable suggestion. Although our study did not find significant variations in suicidal ideation across different residency years, we acknowledge the relevance of this aspect. Accordingly, we have supplemented our manuscript with a concise discussion on the cultural dynamics and inter-level relationships among medical residents.

Please refer to Lines 303-315 on Page 20.

 3- A comparison of the number of working hours scheduled for medical residents in Iran to international standards could be beneficial to this discussion. 

In response to your valuable feedback, we have now included a comparison of the working hours of medical residents in Iran with international standards in our discussion.

Please refer to Lines 293-303 on Page 20.

 4- Address the external validity of the sampling method's results for medical specialty trainees in Tehran universities. 

Thank you for your comment. We have addressed this concern in our discussion as a limitation of our study.

Please refer to Lines 342-346 on Page 22.

 5- Please discuss the statistical power for some of the mentioned results, such as the non-significant disparity in the risk of suicide among different residency fields, etc 

Following your suggestion, we have expanded our discussion to include the statistical power for certain results, such as the lack of significant differences in suicide risk across various residency fields.

Please refer to Lines 315-320 on Page 20 & 21.

 6- Some parts of the discussion are out of the scope of the results of this study. Such as “COVID-19-related factors also affected the prevalence of suicidal thoughts among medical residents”. What is the role of other factors? Such as socio-economic status, timely access to most efficient Covid-19 vaccines etc. 

We apologize for any unclear wording. By the phrase "COVID-19 related factor" in this section, we mean the factor " Death of Close Persons because of COVID-19," which was significantly linked to suicidal thoughts, as detailed in our results section. We have now revised the relevant sentence in the discussion section to eliminate any ambiguity.

Please refer to Lines 321 & 322 on Page 21.

 One step in measuring suicidal ideation among doctors is conducting institutional reviews. With the three prominent medical universities with residency programs in Tehran, an institutional review should be done to have a more comprehensive picture of this subject. There are Medical Students' Union Councils and even some associations of medical specialty residents in each university, and the authors should get their representatives' opinions in this regard for having a comprehensive study. It is also beneficial to suggest these for future research. 

Thank you. We have now incorporated a recommendation for further research involving institutional reviews.

Please refer to Lines 352-356 on Page 22.

 

Reviewer #2

1. What was the target population? How many medical residents did you sample from? 

The study's target population encompassed all medical residents from five Tehran universities: Tehran University of Medical Sciences (TUMS), Shahid Beheshti University of Medical Sciences (SBUMS), Iran University of Medical Sciences (IUMS), AJA University of Medical Sciences, and University of Social Welfare and Rehabilitation Sciences (USWR), covering various stages of residency training. According to recent information from the Iranian Ministry of Health, and Medical Education, these universities together accommodate around 4,000 medical residents, which accounts for about one-third of the nation's total number of medical residents.

 The minimum sample size was calculated using the following formula:

n=N−1d2+z1−α/22p(1−p) Nz1−α/22p(1−p)

where α=0.05, z1−α/2=1.96 is the 100th percentile of the standard normal distribution for a 95% confidence level, p=0.5 is the proportion of residents with suicidal thoughts (since no previous study has been done, we assumed the highest possible value of 50% to obtain the largest sample size), d=0.05 is the margin of error, and N=4000 is the total number of residents working in health care centers in Tehran. Based on this formula, the minimum sample size was 351.

Thank you for your input. We have incorporated details about our target population into the Methods section as suggested.

Please refer to Lines 115-118 on Page 7 & 8.

2. Under study design, you state that the data collection period was from 1 April to 10 March 2022, between the third and fourth waves of the COVID-19 pandemic in Iran. This is not clear. Was it April of 2022 or 2021? We apologize for the error in the text. The study's time frame has been corrected to indicate that the survey was available 24/7 from Jan 1, 2022, to Feb 28, 2022, and concluded upon reaching the desired sample size.

Please refer to Lines 108 & 109 on Page 7.

3. The conclusions do not seem to be connected to the study findings. Please rewrite them. 

The conclusions have been revised to accurately reflect the findings of the study.

Please refer to Lines 358-364 on Page 23.

4. Explain why you chose to use linear regression for analysis for categorical variables. 

In our study, we utilized linear regression to analyze the continuous score of suicidal ideations as the dependent variable, despite dealing with other categorical variables. This methodological choice was made to effectively quantify the influence of various predictors on suicidal ideation, converting categorical variables into a numeric format for comprehensive analysis. Employing linear regression was advantageous for its ability to adjust for potential confounders and to simultaneously examine multiple variables.

5. In results, you state that sex, marital status, specialty, year of residency, having children, living arrangement, accommodation, history of COVID-19 infection, and history of admission due to COVID-19 infection were not significantly associated with suicidal ideation (P > .05 for all variables). Please specify that you are referring to statistical significancy. 

We have now revised the mentioned sentences in our manuscript to explicitly clarify that we are referring to statistical significance.

Please refer to Lines 221-224 on Page 16.

6. You collected data on death of a relative due to COVID-19. Did you also get information on death of a relative due to other causes? 

In response to your inquiry, we focused exclusively on collecting data regarding the death of relatives due to COVID-19 and did not include information on deaths from other causes.

7. Were there any statistically significant differences across specialties? You mentioned many but they are not reflected in the results. 

We have now revised the results section to explicitly state that our analysis found no statistically significant differences in suicidal ideation across different medical specialties.

Please refer to Lines 221 & 224 on Page 16.

8. What has this study contributed to the already existing literature. Please state/describe it. 

Thank you for highlighting this important aspect. We have addressed this issue in the Introduction and Conclusions sections of our study.

Please refer to Lines 81-95 on Page 6, and Lines 358-364 on Page 23.

9. Were there any differences across the different schools/universities? Please present this. 

In response to your query regarding differences across various schools/universities, this aspect was not a focus of our study and hence was not analyzed. However, we recognize the potential value of this line of inquiry and will consider it for future research

10. Please reformat your tables to fit on one page each. 

We attempted to reformat the tables by minimizing the font size, modifying the spacing, and rearranging the columns and rows. However, given the extensive data in Tables 1 and 2, fitting them onto a single page was impractical. To enhance readability, we have repeated the table headers across pages.

---

## [Decision Letter · Decision Letter 1]

5 Feb 2024

PONE-D-23-19377R1Factors associated with suicidal ideation among medical residents in Tehran during the COVID-19 pandemic: a multicentric cross-sectional surveyPLOS ONE

Dear Dr. Kamalzadeh,

Thank you for submitting your manuscript to PLOS ONE. After careful consideration, we feel that it has merit but does not fully meet PLOS ONE’s publication criteria as it currently stands. Therefore, we invite you to submit a revised version of the manuscript that addresses the points raised during the review process.

The authors have adequately addressed most of the previous review comments. It would be nice to address the following remaining concerns.

1. The authors did not confirm the authenticity of the respondents. This should be acknowledged as a weakness to this study in the limitations section.

2. In the calculation of the sample size, it was not clear if the assumed 50% was referring to suicidal ideation or attempt or both ideation and attempt. It was also not clear if the 50% referred to low or high risk of suicide. This needs to be clarified.

3. When addressing the comment on confounding factors, the authors stated that they omitted additional factors. These need to be indicated in the revised manuscript.

We look forward to receiving your revised manuscript.

Kind regards,

Juan Jesús García-Iglesias, Ph.D.

Academic Editor

PLOS ONE

Journal Requirements:

Additional Editor Comments:

The authors have adequately addressed most of the previous review comments. It would be nice to address the following remaining concerns.

1. The authors did not confirm the authenticity of the respondents. This should be acknowledged as a weakness to this study in the limitations section.

2. In the calculation of the sample size, it was not clear if the assumed 50% was referring to suicidal ideation or attempt or both ideation and attempt. It was also not clear if the 50% referred to low or high risk of suicide. This needs to be clarified.

3. When addressing the comment on confounding factors, the authors stated that they omitted additional factors. These need to be indicated in the revised manuscript.

Reviewers' comments:

Reviewer's Responses to Questions

**Comments to the Author**

1. If the authors have adequately addressed your comments raised in a previous round of review and you feel that this manuscript is now acceptable for publication, you may indicate that here to bypass the “Comments to the Author” section, enter your conflict of interest statement in the “Confidential to Editor” section, and submit your "Accept" recommendation.

Reviewer #1: All comments have been addressed

Reviewer #2: (No Response)

2. Is the manuscript technically sound, and do the data support the conclusions?

Reviewer #1: Yes

Reviewer #2: Yes

3. Has the statistical analysis been performed appropriately and rigorously? 

Reviewer #1: Yes

Reviewer #2: Yes

4. Have the authors made all data underlying the findings in their manuscript fully available?

Reviewer #1: No

Reviewer #2: Yes

5. Is the manuscript presented in an intelligible fashion and written in standard English?

Reviewer #1: Yes

Reviewer #2: Yes

6. Review Comments to the Author

Reviewer #1: (No Response)

Reviewer #2: The authors have adequately addressed most of the previous review comments. It would be nice to address the following remaining concerns.

1. The authors did not confirm the authenticity of the respondents. This should be acknowledged as a weakness to this study in the limitations section.

2. In the calculation of the sample size, it was not clear if the assumed 50% was referring to suicidal ideation or attempt or both ideation and attempt. It was also not clear if the 50% referred to low or high risk of suicide. This needs to be clarified.

3. When addressing the comment on confounding factors, the authors stated that they omitted additional factors. These need to be indicated in the revised manuscript.

7. PLOS authors have the option to publish the peer review history of their article (what does this mean?). If published, this will include your full peer review and any attached files.

Reviewer #1: No

Reviewer #2: **Yes: **Dr. Godfrey Zari Rukundo

---

## [Author Response · Author response to Decision Letter 1]

25 Feb 2024

1. The authors did not confirm the authenticity of the respondents. This should be acknowledged as a weakness to this study in the limitations section. We have now addressed the concern regarding respondent authenticity in the limitations section, lines 354-356 on page 22.

2. In the calculation of the sample size, it was not clear if the assumed 50% was referring to suicidal ideation or attempt or both ideation and attempt. It was also not clear if the 50% referred to low or high risk of suicide. This needs to be clarified. We have clarified the assumption of the 50% prevalence rate as referring to suicidal ideation, not attempts, covering both low and high risk of suicide, on page 8, lines 130-132.

3. When addressing the comment on confounding factors, the authors stated that they omitted additional factors. These need to be indicated in the revised manuscript. We have included the omitted confounding factors in the revised manuscript, specifically addressed on page 22, lines 348-353.

---

## [Editor Report · Decision Letter 2]

27 Feb 2024

Factors associated with suicidal ideation among medical residents in Tehran during the COVID-19 pandemic: a multicentric cross-sectional survey

PONE-D-23-19377R2

Dear Dr. Kamalzadeh,

We’re pleased to inform you that your manuscript has been judged scientifically suitable for publication and will be formally accepted for publication once it meets all outstanding technical requirements.

Kind regards,

Juan Jesús García-Iglesias, Ph.D.

Academic Editor

PLOS ONE

Additional Editor Comments (optional):

I have reviewed the changes made and I am pleased to see that you have addressed the comments and suggestions in a very satisfactory manner. The manuscript has significantly improved.
---

## [Editor Report · Acceptance letter]

7 Mar 2024

PONE-D-23-19377R2 

PLOS ONE

Dear Dr. Kamalzadeh, 

I'm pleased to inform you that your manuscript has been deemed suitable for publication in PLOS ONE. Congratulations! Your manuscript is now being handed over to our production team.

Kind regards, 

on behalf of

Dr. Juan Jesús García-Iglesias 

Academic Editor

PLOS ONE